# Active Lifestyle and Mobility of Adults with Vision Impairment: A Multiphase Mixed-Methods Study

**DOI:** 10.3390/ijerph20196839

**Published:** 2023-09-27

**Authors:** Deborah Barasche-Berdah, Ora Paltiel, Raanan Raz

**Affiliations:** Braun School of Public Health and Community Medicine, Hadassah Medical Organization, Faculty of Medicine, Hebrew University of Jerusalem, POB 12272, Jerusalem 92210, Israel; orap@hadassah.org.il (O.P.); raanan.raz@mail.huji.ac.il (R.R.)

**Keywords:** vision impairment, health disparities, active lifestyle, spatial data, qualitative methods

## Abstract

Individuals with vision impairment (VI) are less physically active than their sighted peers, heightening their risk of chronic illness. This study aimed to explore real and perceived barriers and promoters of an active lifestyle among adults with VI. We used mixed-methods, including a quantitative analysis of spatial data and a qualitative analysis of input from focus groups. The data were analyzed using descriptive statistics, graphical and thematic analysis. The spatial analysis highlighted frequent impediments, including high proportions of inaccessible crosswalks and stairs, a high density of obstacles (12/km) and almost inexistent guidance markings (0.1/km). Factors influencing active lifestyle of individuals with VI reported in the focus groups included: VI severity and self-confidence; accessibility of the physical environment as well as support and consideration of the society; use of behavioral strategies, striving for good health and willpower. Combined psychosocial, behavioral and infrastructural modifications could enhance active lifestyles among adults with VI. Consideration of the needs of individuals with VI is critical among environmental planners. Sustainable solutions for improving accessibility and mobility in the city for individuals with VI will influence not only their social integration but will enable them to adopt an active lifestyle and reduce their risk of chronic illness.

## 1. Introduction

Vision impairment (VI) is a global issue, affecting at least 2.2 billion people around the world [1]. In Israel, approximately 24,000 individuals are classified as blind or with VI [2]. Individuals with various levels of impaired vision, from mild VI to complete blindness, have poorer general health than sighted population [3,4]. They are also more likely to face difficulties in mobility and function related to quality of life [5,6]. Moreover, several studies have shown that individuals with VI are less physically active than their sighted peers [7,8,9,10,11,12].

Regular engagement in physical activity (PA) and adoption of an active lifestyle reduce the risk of cardiovascular disease and chronic conditions such as obesity, high blood pressure and diabetes [13,14]. In addition, PA positively influences well-being and psychological health [15,16]. Despite the evidence that individuals with VI are less physically active and in poorer physical condition, few relevant interventions for this population have been studied [17].

Accessibility, disabling physical and social environment, organizational opportunities, transportation, lack of information, personal safety, and health behaviors are among the factors that influence active lifestyle among specific populations with vision impairment [18,19,20,21,22]. Exercise restriction due to fear of falls is very common in adults with VI [23,24,25]. Furthermore, individuals with VI were found to have limitations in mobility [26], and a tendency to leave their home less frequently [27], which may affect their ability to integrate PA into their daily routines. Environmental factors may pose barriers to active lifestyle, thereby producing disabling situations and preventing equitable participation and PA by all citizens [28]. The urban environment poses a special challenge. Jerusalem is an ancient city, a beautiful fusion of old and new, expressed in different forms of architecture. It is very crowded, with a challenging, hilly topography. Jerusalem is a walled city and stands on hills at an elevation of 785 m. Most of the structures are covered in Jerusalem stone and some of the streets are cobble-stoned, preserving the historical look of the city. In the heart of Jerusalem lies the old city. The old city includes numerous historical monuments, religious sites and narrow and bustling alleyways. In contrast, the newer parts of Jerusalem illustrate the combination of modern architecture with restoration projects. This complex urban environment may influence mobility and activity in the city for individuals with VI.

The purpose of this study was to describe perceived and real barriers and promoters of active lifestyle among adults with VI in an urban setting.

## 2. Materials and Methods

This multiphase mixed-methods study combines a quantitative analysis of spatial data from “real life” routes of individuals with VI (direct observation phase) and a qualitative sub-study. The participants were recruited from three local vision loss organizations, directly at the sites with the help of the coordinators and/or directors of the organizations. We identified these organizations as physical spaces being used by our target group. The participation in the study was voluntary. We used a purposive sampling strategy with variability regarding the participants’ age, ethnicity, VI severity, neighborhood, and PA level. Descriptive data were collected from all participants. The inclusion criteria for participation in the study were: (a) age ≥ 18 years, (b) residents of the greater Jerusalem area, (c) native language speakers (Hebrew) and (d) self-report of visual acuity worse than 6/12 [29], or classified as blind by the State of Israel. We excluded participants with a cognitive or a mobility impairment or communication disorder. We conducted the qualitative study first, then the route documentation phase. Five subjects participated both in the quantitative and qualitative phases, 17 participated in the qualitative phase only and five in the quantitative phase only. The same recruitment methods and inclusion criteria were used for both the quantitative and qualitative phases. The Institutional Review Board approved this study and all participants provided written or recorded informed consent (approval number: 16072018).

### 2.1. Quantitative Phase

One of the researchers accompanied ten individuals with VI from different neighborhoods on their routine walking routes, from their home to destinations such as the closest public transportation station, food store or workplace. Using the Collector for ArcGIS© application, we configured the fields for data collection requirements, documented and mapped the route, and further documented relevant entities, such as obstacles (We usually used the term ‘obstacle’ for physical obstacles reported by participants. The term ‘barrier’ was related to a more general category of ‘accessibility barriers’) and specific infrastructure entities for individuals with VI. We then documented geographic location, descriptive data, and photographs of these features. We defined prior to data collection, according to literature review and focus groups (FGs) conclusions, which entities to document systematically. Thus, we documented along the routes all the crosswalks (full accessibility was assessed by the presence of both audible signal and tactile paving; partial accessibility with one of these; inaccessibility if both were absent), public transportation stations (accessibility was assessed by the presence of audible signals), stairs (accessibility was assessed by the presence of visual/tactile warning strips, and we also documented the presence of ramps), narrow and unsmooth sidewalks and proximity to construction sites. We also documented the presence of guard rails, guidance marking (tactile paving/paths) and public fitness facilities. The physical sidewalk obstacles reported by participants in real-time were grouped into one category and denoted as “obstacles-other”. This included documentation of problematic locations of electric polls, lampposts, garbage cans, vegetation, mail boxes, street or store furniture as well as inexistent sidewalks, proximity to traffic, poor lighting or dangerous descents. The nature of accessibility of each documented entity was assessed according to predefined coding rules agreed upon by all researchers (presence of audible signals, tactile paving or visual warning strips) and participants’ reports (defined as the “Obstacles-other” category). We performed a separate analysis of the accessibility of crosswalks, public transportation stations and stairs. We then divided the different entities into “accessibility barriers” and “accessibility arrangements”. The elements related to physical obstacles or inaccessibility issues were denoted as “accessibility barriers” and included the categories of “Obstacles-other”, inaccessible crosswalks/public transportation stations/stairs, unsmooth and narrow sidewalks and disturbing construction sites (due to lack of safety fence or safe walking spaces). Elements defined as accessible or as promoters of active lifestyle for individuals with visual impairment were designated as “accessibility arrangements” and included accessible crosswalks (with full and partial accessibility), accessible public transportation stations, accessible stairs, as well as “other accessibility arrangements”, which included guardrails, guidance marking, and public fitness facilities along the routes (Appendix A. Description of the documented Entities).

To obtain a descriptive and comparative index, the density per km was calculated for each category of documented entities (number of entities divided by the total walking distance of all routes in km). We used descriptive statistics and graphical analysis to describe the data features.

The limited sample size precludes a valid sensitivity analysis.

### 2.2. Qualitative Phase

We assembled six FGs (n = 22). Prior to recruitment, we conducted a pilot study with three participants, in order to ensure that the questions worked in practice. An interview guide was used to facilitate discussion. Qualitative research experts reviewed the questions prior to the FGs and consequently to the pilot study. The FGs took place at the accessible local vision loss organizations. The discussions were audio-recorded, and transcribed verbatim. We analyzed the qualitative data using the COM-B model (Capabilities, Opportunities, Motivations and Behavior) as a framework for data interpretation [30]. This model can be used to structure an analysis into barriers and promoters for change behavior [31]. The analysis process was both inductive (coding derived from the data) and deductive (driven by the conceptual categories of the COM-B framework). As a first step, we performed an initial coding of the transcribed data; we then identified categories and key themes. One of the researchers met frequently with an independent qualitative research expert to discuss the coding frame throughout the data analysis process.

## 3. Results

### 3.1. Quantitative Phase

Ten individuals with VI from different neighborhoods in the city (Beit vegan, Rehavia, Talpiot, Arnona, Katamon, Kiryat Hayovel, Har Atsofim, Armon Anatsiv, Talbiya, Givat Shaul) participated in the route documentation phase and 33 routine routes were documented (Table 1, Appendix A). A total of 556 entities were documented along the routes (Figure 1, Appendix A), with a high proportion of obstacles (35%).

Among the 426 entities (77% of all documented entities) categorized as “accessibility barriers”, 45% were “obstacles-other”, (Figure 2).

In addition, 130 entities were categorized as “accessibility arrangements” (23% of all documented entities), (Figure 3).

A more detailed analysis was performed for crosswalks, public transportation stations and stairs. Among all the documented crosswalks (n = 137), 41% were inaccessible (without tactile paving and audible signals), another 38% were partially accessible (tactile paving only), and 21% fully accessible (tactile paving and audible signals). We found 60% of the documented public transportation stations (n = 53 in total) to be accessible (with audible signals) for individuals with VI. The documented stairs (n = 44) were mostly inaccessible (95%), lacking any visual or tactile warning strips.

The density of accessibility barriers was 26.5/km, compared with only 8/km accessibility arrangements and with a 12/km density for obstacles. We found a low density of guidance marking intended for individuals with VI (0.1/km) (Figure 4).

Selected photographs of the documented entities taken along the routes are presented in Figure 5 and Figure 6.

### 3.2. Qualitative Phase

Twenty-two individuals with VI participated in FGs (Table 2). We present our findings according to the constructs of the COM-B model.

#### 3.2.1. Capabilities

Physical capability;

Physiological function: VI severity and other health conditions;

The severity of the VI is a crucial factor for adopting active lifestyle. According to our respondents, the more severe the VI, the more significantly it will affect an active lifestyle adoption. When some participants described a manageable situation concerning mobility and PA level, they justified it thanks to the residual vision they have: “*There is a wide spectrum of VI, I see relatively well compared to other individuals with VI and I can do PA which doesn’t necessarily need to be adapted for individuals with VI*”;

Participants frequently compared each other in terms of VI severity or compared their habits concerning PA before and after their vision loss *(“Now, I’m not so active for the simple reason that my vision has deteriorated”)*. The participants also described the specific limitations in mobility or PA that result directly from their disability. They gave examples of limitations in specific types of sport (ball and team games) and described their ‘urban challenges’ with public transportation or crosswalks, and their difficulties to ‘navigate’ in the city because of the VI. Some participants also described health conditions as an additional barrier to active lifestyle, such as back pain, or age-related issues;

Psychological capability;

Lack of self-confidence;

Most of the respondents described their conflict between the desire to have an active lifestyle and their lack of confidence and vulnerability due to their fear to fall or being hurt: *“I fell down in the past and it was traumatic”; “I walk very slowly because of the fear of falls. I have fallen several times already…I am scared to walk on the sidewalk, I am scared to death”.*

#### 3.2.2. Opportunities

Physical environment

Accessibility along routes

Accessibility was brought up as a major concern. The respondents suggest that improving the physical environment to be more accessible and inclusive is a way to encourage mobility and activity. Thus, the participants recognized the authorities’ efforts and several improvements in the city infrastructure accessibility over the years. However, they also mentioned many inaccessibility issues in the city and described the lack of enabling physical environment as a major barrier for active lifestyle adoption. They reported the lack of markings for individuals with VI including the absence of accessible signage in public spaces and the missing guidance of visual/tactile paths (“*Inappropriate signage means inability to walk for individuals with VI*”). Participants also reported the importance of voice-based assistance in the city environment and complained about their inefficiency, inadequacy, or absence. Furthermore, participants addressed the many physical obstacles they encounter along the routes they take in their daily routine: “*I live on a street with almost no sidewalk left because of the parked cars. I regularly must walk on the road...I endanger myself and I endanger the drivers also. Mobility, which is an integral part of adopting active lifestyle, is problematic here*”. However, the physical obstacles or inaccessibility partly stem from non-compliance with the law. The participants referred to disobedience regarding the law that sometimes explains the lack of accessibility: “*Before the consideration that we are asking for, before the special care, they have to respect the law! It will solve 90% of the problem. For example, if the voice announcement in the bus was working as the law requires, or if the drivers were not illegally parking their cars on the sidewalks…Law enforcement is also problematic*”.

Accessibility of transportation

In all FGs, transportation in the city was mentioned as problematic and inaccessible. The participants complained about the inefficient and non-adapted voice assistance (“*The button for audio announcement in the station is so tiny, it’s not a big, colored and noticeable button. You must almost ‘embrace’ the information board to find it”; “Sometimes the audio announcements suddenly stop in the bus. I don’t know where I am, which station it is, I get lost*”), inadequate signage (“*In the bus stations you can hardly find signage in Braille or in large letters*”), and sometimes the lack of consideration of bus drivers (“*The bus drivers do not stop at the bus station, they stop far away, and I don’t see. It is the new trend today; the bus drivers do not stop at the station and I am with a white cane*”).

Opportunities for PA

The sport center and type of activity must be accessible. The participants referred to a sport center located at the “School for the Blind” that offers an adapted environment for PA. “*I only go there to do sports; I don’t think that there is another center like this in the whole country. I come here to swim, and I exercise in the gym. It makes me leave the house, it’s the only thing I do in terms of PA*”. In addition to physical accessibility, the gym was also financially accessible: “*The gym in the school for blind is almost free for us, it costs 10 NIS per year, and the pool is completely free… the sport center here is very helpful both in terms of price and of physical accessibility.*” Safe and adapted sport or activity was also important for participants: “*What is good here for the blind people is walking on the treadmill in the gym because for us, walking in the street means confronting obstacles. Here, you can walk faster thanks to the machine, there is no danger*”.

Social environment

The importance of social support and employment status

The status and inclusion of individuals with VI into societal life were discussed during the discussions. Social support based on communication and assistance was described as a facilitator for active lifestyle. Encouragement and help of friends, peers, family, mobility trainer or even strangers stimulated participants to be more active: “*Here (Jerusalem), the people understand much more what it means to be blind. For example, my husband is also visually impaired and uses a white cane. In the street, people immediately come to help*”; “*If I am going to the gym, it is thanks to my friend E., he constantly encourages me to go*”. In contrast, loneliness and limited social relationships altered their activity: “*I don’t have so many friends… so I stay home, in my familiar environment…I only have two friends and very often they don’t want to go out*.”

Employment was also mentioned as a factor influencing adoption of an active lifestyle. The workers described employment as providing them with an important motivation for leaving home and being active. In contrast, unemployment justified their inactivity: “*Since I quit my job about 5 years ago my life is not so active*.”

Social consideration and public awareness to the disability

Positive and inclusive attitudes of sighted persons was described as an important motive for active lifestyle, but their indifference acts a major barrier. The participants described the lack of consideration of pedestrians, car and bus drivers or service providers: “*You have to be so careful in the streets and several times there are such absurd things…Garbage cans for example in the middle of the sidewalk…We need more consideration, more respect”; “ Nobody pays attention to blind people*”.

The participants also described the lack of awareness of sighted people to individuals with VI: “*If they see someone in a wheelchair they will stop, help, open the door. But to notice that a person is visually impaired is more difficult*”; “*People do not know what a white cane is. They do not understand. Someone even asked me ‘What is it? A camera?...We need to create more awareness about VI among sighted people…People are walking in the street and they are expecting from the person in front to move aside, no matter if he is blind. They are walking while looking at their phones and they do not pay attention to the blind people*”.

#### 3.2.3. Motivation

Reflective motivation

Use of behavioral strategies

The participants described special strategies and skills as motivators for an active lifestyle. For instance, mobility in familiar environments: “*It’s a familiar and well—known place. I have been coming here for eight years so it is much easier for me because I remember; I know every crosswalk nearby, all the buses. I know the routes from here to a lot of locations*”. Another strategy is the use of aids such as a white cane, low vision devices or a guide dog: “*The combination of my biggest love which is dogs and the need of outdoor assistance actually fit together. The dog leads me in the best ways. If she doesn’t progress, it’s a sign that there is an obstacle and I listen to her*.” Furthermore, the respondents described help from sighted people as a strategic issue to facilitate active mobility: “*I like to follow people, I walk behind them and copy their steps*”. However, when lacking, this dependence on others for help was also described by most participants as a barrier: “*I will not walk alone. It limits me. I depend on someone… I believe it is one of the barriers of individuals with VI, most of them do not leave the house without an escort*.”

Striving for good health

The desire to maintain good health was mentioned by some respondents as a motivator for active lifestyle: “*The elevator is for your comfort and the stairs are for your health”; “We know that being physically active is very healthy, it is healthy for the body and for the mind*”.

Automatic motivation

Willpower

The impact of willpower was mentioned: “*I don’t accept the fact that because somebody is visually impaired or blind, he has to give up. Willpower will help him to be active*”. On the other hand, several participants mentioned the negative impact of lack of willpower to be more active and the resulting sedentary behavior, as one participant confesses: “*All the reasons I will give you are excuses. I’m just a person who does not want to be active…It’s not that I can’t, I am just lazy*”.

## 4. Discussion

In this mixed methods study, the findings highlight several real and perceived barriers and promoters to capabilities, opportunities, and motivations for an active lifestyle among individuals with VI. The spatial investigation confirmed the qualitative analysis conclusions and highlighted several issues in the physical and social environments (Opportunities). However, alongside the consistency during FGs regarding the importance of these factors, there is an interpersonal physical and psychological variability, which also has an influence (Capabilities). Adaptive strategies, increased willpower and raising awareness about the benefits of PA (Motivations) need also to be promoted and will strengthen behavior change and adoption of an active lifestyle among individuals with VI.

Results of the route documentation strengthen the findings from the qualitative sub-study and illustrate the daily mobility challenges of individuals with VI in an urban environment. Our findings highlighted a high density of physical obstacles, guidance markings almost inexistent, or a high proportion of inaccessible crosswalks and stairs. Most of the physical obstacles documented along the routes were also mentioned in the FGs. Participants complained about their difficulties with stairs and lack of warning marking, and we found 95% of stairs to be inaccessible. We also documented an almost non-existent density of marking intended for individuals with VI (0.1/km), supporting the claims made in the FGs. Moreover, the high density of obstacles documented along the routes (12/km) may explain the despair of the participants regarding navigation in the city. A study, similar to our quantitative phase, focused on community barriers to PA experienced by people with motor or vision impairments. The study was conducted in New York, combining quantitative and qualitative methods and including both individuals with VI and sighted subjects [32]. Problems with sidewalk pavement and puddles resulting from poor drainage were the most common environmental barriers mentioned by individuals with VI. Our use of objective measures to quantify environmental barriers in daily routes of individuals with VI probably enabled us to document additional accessibility-related features and entities. Another relevant study focused on pedestrian behavior in roundabouts and compared seeing people’s behavior to that of individuals with VI. The latter waited three times longer to cross compared with sighted people and 6% of the crossing attempts of individuals with VI were judged as dangerous and required intervention, a phenomenon that was not observed in the control group [33]. Therefore, accessible crosswalks with auditory output are a reliable resource for individuals with VI, for locating traffic lights in their routes and announcing when to cross safely, but also as auditory cues for orientation in the city [34]. These findings emphasize the importance of crosswalk inaccessibility, which was the norm in the routes we have documented, as 79% of the documented crosswalks lacked an audible signal. When the audible cues don’t exist, the individuals with VI still need to rely on other indicators as environmental, visual and tactile clues. All these issues must be considered when developing an accessible crosswalk. According to our findings, improving accessibility in the city is necessary, and efficient transportation must be provided. Besides, social advocacy is essential, including improving support and consideration for their condition. The data on parked cars, uncut vegetation, store furniture or garbage cans on the sidewalk illustrate the lack of awareness and/or consideration of the sighted population mentioned in the FGs, highlighting the social aspect of their mobility in the city.

Our findings are in line with other studies that investigated the experience of individuals with VI in relation to health and daily life functioning. In a qualitative study that examined how older individuals with VI manage their general health, the need for support groups and assistance and the importance of social support in coping with daily challenges were presented [35]. Another study described the supports and barriers in daily life of individuals with vision loss due to diabetes, and highlighted the impact of the physical (“the world isn’t accessible”) and social (peer interaction, community settings) environments [36]. The importance of a supportive social environment was also evident in a study from Ireland, which identified attitudes of sighted persons as one of the main barriers to the use of public transportation among individuals with VI [37].

Our findings regarding the importance of psychological factors and opportunities for PA were also highlighted in previous qualitative studies among blind people [19,38]. Similarly, some of the behavioral strategies influencing mobility noted in our study have also been raised in previous studies on navigation behaviors, such as the importance of navigation training, assistive technologies and mobility in familiar environments [39,40].

Future interventions and research may examine a hypothetical model that emerges from the findings, and that should be based on different approaches. An environmental approach will be based mainly on adapting the physical environment (accessibility of activity centers and urban infrastructures, implementation of accessibility legislation, cooperation with local authorities, community organizations and government ministries). A social approach may focus on social advocacy that will reinforce support, accompaniment, raising awareness and consideration regarding the disability (social media campaign, community-based interventions, and inclusive sports with seeing persons). A behavioral approach may focus on teaching coping strategies and special skills for effective physical activity and mobility (adapted sports activities, mobility training). Finally, a psychological approach should focus on psychological training for self-confidence and motivation (lectures and workshops, motivational interviews).

There are some limitations to this study. First, in the quantitative phase, we did not document routes and elements from the entire city. The descriptive data collected from each participant (as PA assessment) was based on self-report. For some of the documentation, based on participant’s report, their assessment of hazard may depend on VI severity or level of autonomy. However, this is also one of the research strengths, and reflects the participant’s routine and their perspective regarding challenges in mobility. The number of participants was small, and the sample was not necessarily representative. It also included only participants from Jerusalem, a unique crowded city with a hilly topography, limiting the ability to generalize the results. Furthermore, we acknowledge that more isolated individuals or those who do not use the services of the selected centers were probably not represented in our sample. However, for qualitative studies, limited variance is preferable to avoid overestimation of the importance of extreme observations [41]. This more homogenous sample avoids complexification based on extreme cases (in our study, very isolated individuals with VI, or independent and socially integrated with their sighted peers).

## 5. Conclusions

This descriptive study used a multi-phase mixed research methods and theoretical framework to obtain a deeper understanding of the objective and perceived factors influencing adoption of an active lifestyle among individuals with VI. These observations provide information about some of the adjustments required for people with VI to increase their PA and improve mobility.

The findings emphasize the interplay between the physical urban and social environment and their relationship with the physical and psychological factors in influencing PA among individuals with VI. There may be a need to invest in teaching individuals with VI new skills and coping strategies in navigation in both the physical and the social environments, including learning how to interact with the seeing population. Furthermore, we found that despite existing legislation and accommodations, there are still deficiencies in accessibility in the urban environment. Consideration of the needs of individuals with VI is critical among environmental planners in order to create a physical environment that promotes PA and mobility. Despite the limited sample size, we believe that our results are relevant to other individuals suffering from similar impairments. Given current technologies, further similar research could be performed on a mass scale, yielding even greater insights.

Sustainable solutions for improving accessibility and mobility in the city for individuals with VI will influence not only their social integration but will enable them to adopt an active lifestyle and reduce their risk of chronic illness.

Investigating the factors affecting the lifestyle of individuals with VI and raising awareness about their special needs are fundamental for reducing the observed health gaps.

## Figures and Tables

**Figure 1 ijerph-20-06839-f001:**
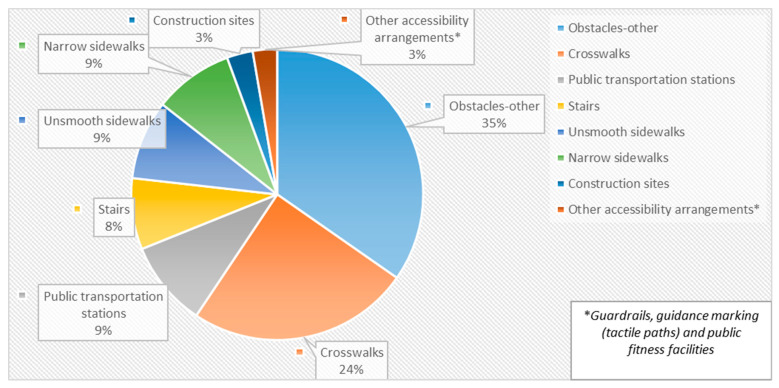
Distribution of Entities in Daily Walking Routes by Entity Category.

**Figure 2 ijerph-20-06839-f002:**
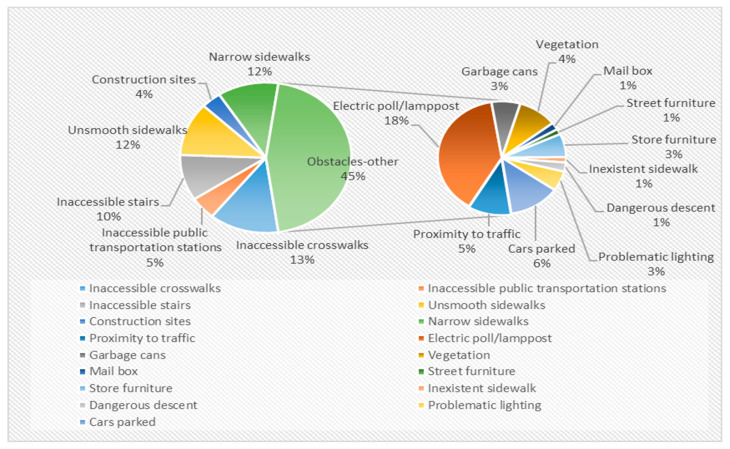
Distribution of “Accessibility Barrier” Entities in Daily Walking Routes.

**Figure 3 ijerph-20-06839-f003:**
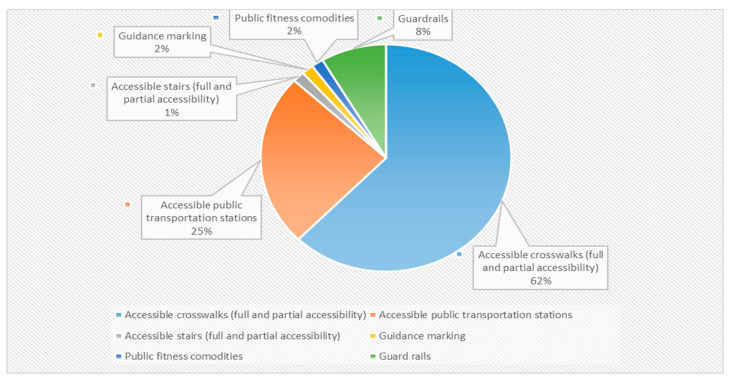
Distribution of “Accessibility Arrangement” Entities in Daily Walking Routes.

**Figure 4 ijerph-20-06839-f004:**
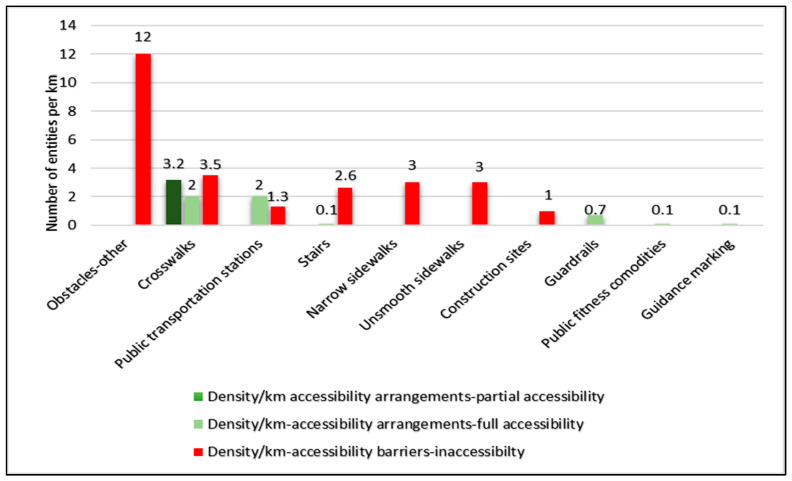
Density per Km for all entities in daily walking routes by category and accessibility criteria.

**Figure 5 ijerph-20-06839-f005:**
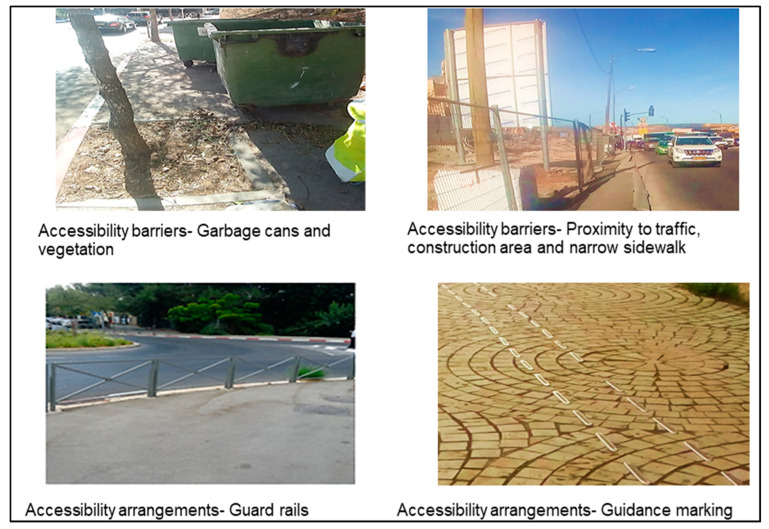
Accessibility Barriers and Arrangements Documentation in Daily Walking Routes-Obstacles and Warnings.

**Figure 6 ijerph-20-06839-f006:**
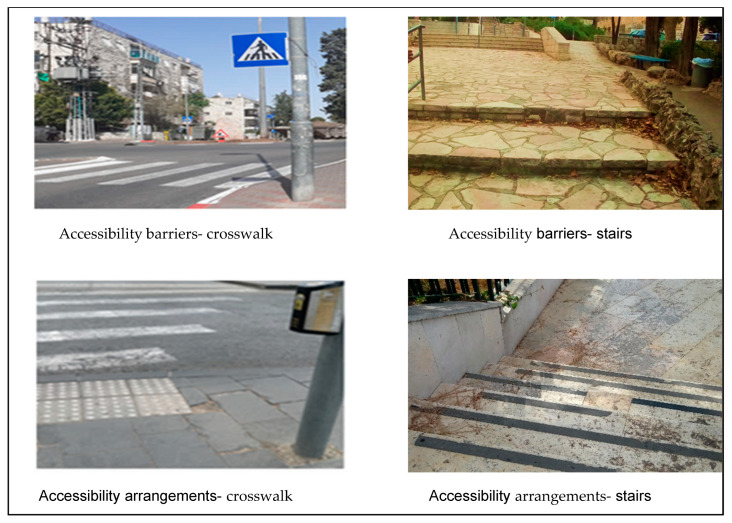
Accessibility Barriers and Arrangements Documentation in Daily Walking Routes-Crosswalks and Stairs.

**Table 1 ijerph-20-06839-t001:** Routes Documentation Details.

Participants	Total Walking Distance (m)	Number of Routes	Route Destinations
Participant 1	1830	4	Bus station, food store, local vision loss organization, place of worship
Participant 2	1359	4	Bus station, workplace, sport center, food store, college
Participant 3	643	3	Food store, place of worship, bus station
Participant 4	1623	3	Bus station, food store, supermarket
Participant 5	1986	5	Bus station (n.1), bus station(n.2), sport center, medical center, supermarket
Participant 6	1470	3	Bus station, food store, shopping center
Participant 7	2500	2	Sport center, local vision loss organization
Participant 8	929	3	Bus station, food store, sport center
Participant 9	2700	3	Sport center, bus station, workplace
Participant 10	1028	3	Bus station, supermarket, tramway station
Total	16,068	33	
Mean	487	3.3	
Standard Deviation	374	0.82	
Maximum	1500	5	
Minimum	48	2	
Median	400	3	
Q1; Q3	230; 539	3; 3.75	

**Table 2 ijerph-20-06839-t002:** Characteristics of Participants in the Qualitative Sub-Study (N = 22).

Participants Characteristics	N (%)
**Gender**	
Men	(36%) 8
Women	14 (64%)
**Marital Status**	
Married	(32%) 7
Single	(45%) 10
Divorced	(14%) 3
Widowed	(9%) 2
**Age**	
Average (Standard Deviation) (Range)	48 (20.17) (18–84)
**Education**	
High School Or Less	(64%) 14
College/University Students	(9%) 2
Ba	(9%) 2
Master	(14%) 3
PhD	(5%) 1
**Occupation Status**	
Employed	(27%) 6
Unemployed	(68%) 15
National Service	(5%) 1
**Blind Certificate**	
Yes	(86%) 19
No	(14%) 3
**Vision Impairment Severity**	
Vision Impairment	(86%) 19
Blindness	(14%) 3
**Causes of Visual Impairment ^1^**	
Retinitis Pigmentosa (RP)	(32%) 7
Glaucoma	(23%) 5
Cataract	(18%) 4
Albinism	(9%) 2
Congenital-Unknown Cause	(9%) 2
Retinal Detachment	(5%) 1
Nystagmus	(5%) 1
Uveitis	(5%) 1
Corneal Conditions	(5%) 1
Retinal Conditions	(5%) 1
**Vision Impairment Duration**	
Less Than 10 Years	(5%) 1
Between 10–30 Years	(36%) 8
More Than 30 Years	(9%) 2
Congenital	(50%) 11
**Assistant Aids In Use ^1^**	
None	(32%) 7
Cane	(23%) 5
Closed-Circuit Television	(23%) 5
Magnifiers	10 (45%)
Guide Dog	(5%) 1
**Weekly Physical Activity (Self-Report) ^2^**	
Less Than 150 Minute	(59%) 13
More Than 150 Minutes	(41%) 9
Average Time (Hours)	2.6
Range	[8; 0]
Standard Deviation	2

^1^ Multiple categories may be documented for one participant; ^2^ Most international guidelines recommend a goal of 150 min/week of physical activity.

## Data Availability

The data presented in this study are available on request from the corresponding author.

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
