# Peer review of "Active Lifestyle and Mobility of Adults with Vision Impairment: A Multiphase Mixed-Methods Study"

_ijerph, 2023, doi:10.3390/ijerph20196839_

Round 1

Reviewer 1 Report

Dear Authors,

The study named 'Active Lifestyle and Mobility of Visually Impaired Adults: Multi-Stage Mixed Method Study' was reviewed. My views on the paper are as follows:

1.     This study aimed to explore real and perceived barriers and  promoters of an active lifestyle among adults with visual impairment . The study was designed in an  mixed-methods, including a quantitative analysis of spatial data and a qualitative analysis of input from focus groups. The conclusion of the study highlights the various real and perceived barriers and incentive modifiers of abilities, opportunities and motivations for active lifestyle among individuals with visual impairment.

2.     The qualitative part of the study was carried out in 10 visually impaired individuals. No information was found about why these individuals were included in the study and where they came from.

3.     The results of the study only reflect the life cycle of these 10 visually impaired individuals. There are similar areas in the life cycles of all individuals, and the study falls into repetition due to the selected individuals.

4.     A sensitivity analysis should have been done for the classifications made in determining the barriers for physical activity stated by the visually impaired in the study. Information should be given on how this assessment was made.

5.     In the study, ‘total walking distance’ and ‘number of routes’ data were shown as mean (SD). However, it would be more informative to give these data with the median and Q1 and Q2.

6.     It would be appropriate to display information on which parts of Jerusalem were included in the study, along with spatial analysis results, on a map via the ArcGIS© application.

7.     No standard acceptable method was used to measure physical activity level. No information is given about the relevant measurement tool. Based on the individual's statement. This situation should be discussed.

8.     Physical activity levels of individuals are closely related to health problems such as diabetes, hypertension, orthopedic and other disability conditions other than visual impairments. It should be evaluated in these co-morbid conditions accompanying visual impairment.

9.      The results of the study include only 10 visually impaired individuals living in a part of a city. Study results cannot be generalized to a city, a country, or other countries.

10.  However, the study is noteworthy as it was made in an ancient city like Jerusalem, as the authors also reported, and it reveals the problems of the disabled. Therefore, the method applied in the study could be improved, it can provide important information to policy makers and administrators for people with disabilities. 

Sincerely Yours

I don't have any comment

Reviewer 2 Report

Thank you for this interesting paper. The introduction in particular is very well written with a very clear rationale for the study. I have a few minor comments to be addressed to make the paper a little bit easier to read and follow.

Firstly, I suggest using the term 'vision impairment' as aligned with the WHO (as you have referenced them). In my experience, patients with VI have indicated a preference with the term 'vision' rather than 'visual' which implies the way they look.

Line 36-73: reduce should be reduces

Line 43: it is unclear what you mean here with 'health managing'

Line 59: You mention 'sight loss' organisations, but I suggest for consistency you use vision loss/blindness instead.

Line 65: What do you mean by 'declined communication skills'?

Methods: How did you determine the sample size for the quantitative study? Especially considering that it less than half the size of the quali component which is very unusual... I suggest including your calculation/justification here.

You speak about obstacles, but only define them in the figures. I would suggest including all of the information that you have in the footers on the table/figures into the methods because it would help to understand your definitions better. Having so many footers is also very unusual in a paper like this. If well described in the methods, then you would only need a heading for the figures.

The figures are quite confusing and it is difficult to tell what the difference between them is... what is an entity? Are they sometimes positive and sometimes negative factors? Other obstacles in figure 1 are 35% but then in figure 2 45%... this links to the previous comment where more description in the methods could help to understand the results better.

Line 110: 23% of entities were accessibility arrangements? How is this calculated considering you state 3% in figure 1?

Line 121: What is the difference between a barrier and an obstacle?

Figure 4: What is 'guidance marking'?

Table 2: not well constructed, I suggest adding another column for the overall category (gender, marital status etc), or horizontal lines between the categories, or align the category name to the left otherwise it looks like there are missing data. Widow should be 'widowed'. Next to Assistant Aids there is some hebrew, can this be translated please?

In the qualitative section some of the quotes are italicised and others not... the reason is unclear but it is easier to differentiate the quotes in italics.

Line 229: 'we have no consideration as blind people'... is this quote correct? It is criticising the behaviour of the blind people which doesn't seem to fit the rest of the section.

The discussion and conclusion seem to give broad recommendations, again speaking about principles rather than specific changes that could be made. It would be useful to have some recommendations on very specific methods to improve this problem e.g. provide transport to a gym setting , etc.

Minor comments as mentioned in the section above.

Reviewer 3 Report

Thank you for requesting I review this paper on a very important topic. This is a great study with a lot of worthwhile information and results. It is also great to see it in the context of Jerusalem. I have a few comments/questions:

Introduction:

Line 31 – not sure grades is the right word perhaps levels of impaired vision is better?

Line 47 – which may affect their ability to integrate PA into their daily routines? Rather than and

Line 49 – ‘The urban environment poses a special challenge’ – in general or specifically Jerusalem as you describe in the next sentence

Line 50 – I think it might be good to add more about Jerusalem and the old and new architecture/urban environment. I would imagine it could be extremely challenging for a VI person. Also perhaps adding in population information etc. for further context may be beneficial.

Materials and Methods

Line 59 – ‘from three local sight loss organizations, directly in the sites’ – are these charities, hospitals? Can you provide any further information?

Line 64 – did you exclude people who could not speak the native language?

Line 65-66 – was there a reason behind conducting the qualitative study first?

Line 66-67 – do you have an IRB approval number?

Line 73-74 – ‘documented relevant entities (such as obstacles, and specific infrastructure entities for individuals with VI)’ – were these as described by the person with VI or assumed by the researcher or both?

Quantitative phase – I am a bit confused by exactly what you did. Did you accompany the VI person on the route and they pointed out the obstacles or did the researcher assume they were obstacles? Then did you go back to the ‘lab’ as such and decide how much of an obstacle something was according to your coding and the participants report? Was the participant prompted to speak about the obstacle or was it of their own volition? Can you provide the coding parameters as a supplementary document? I think you need to carefully rewrite this section and perhaps get someone who isn’t involved in the project to read it to make sure it makes sense. This could be a method which could further this field in the future!

Qualitative phase – was the deductive and inductive analysis conducted in a group? By individual researchers and then compared? Was there a first and second analyser or was it done as a group? And then who was the final say? Was it the qualitative research expert?

Results

Table 2 - (Rp) needs to be changed to RP

3.2.2 – keep the quotes consistent in italics ? or is there a different between the bits in italics and the ones not in italics?

Discussion - Do you think self-reported visual acuity is reliable? Do they get this from doctors or optometrists? May be something to discuss as a weakness.

There a very minor typographical errors and some comments for small changes to the English in the manuscript. 

Round 2

Reviewer 3 Report

Thank you for editing the paper. The addition of methods and supplementary documents makes it much more clear.